# Autophagy Mediates the Degradation of Plant ESCRT Component FREE1 in Response to Iron Deficiency

**DOI:** 10.3390/ijms22168779

**Published:** 2021-08-16

**Authors:** Tianrui Zhang, Zhidan Xiao, Chuanliang Liu, Chao Yang, Jiayi Li, Hongbo Li, Caiji Gao, Wenjin Shen

**Affiliations:** 1Guangdong Provincial Key Laboratory of Biotechnology for Plant Development, School of Life Sciences, South China Normal University, Guangzhou 510631, China; 18728191826@163.com (T.Z.); lclbeyond@126.com (C.L.); 20173029@m.scnu.edu.cn (C.Y.); 20182521047@m.scnu.edu.cn (J.L.); luoshuiwangcan@126.com (H.L.); gaocaiji@m.scnu.edu.cn (C.G.); 2Key Laboratory of South Subtropical Fruit Biology and Genetic Resource Utilization, Ministry of Agriculture and Rural Affairs, Guangdong Provincial Key Laboratory of Tropical and Subtropical Fruit Tree Research, Institute of Fruit Tree Research, Guangdong Academy of Agricultural Sciences, Guangzhou 510640, China; xzd1010@163.com

**Keywords:** protein degradation, autophagy, ESCRT complex, FREE1, iron-deficiency

## Abstract

Multivesicular body (MVB)-mediated endosomal sorting and macroautophagy are the main pathways mediating the transport of cellular components to the vacuole and are essential for maintaining cellular homeostasis. The interplay of these two pathways remains poorly understood in plants. In this study, we show that FYVE DOMAIN PROTEIN REQUIRED FOR ENDOSOMAL SORTING 1 (FREE1), which was previously identified as a plant-specific component of the endosomal sorting complex required for transport (ESCRT), essential for MVB biogenesis and plant growth, can be transported to the vacuole for degradation in response to iron deficiency. The vacuolar transport of ubiquitinated FREE1 protein is mediated by the autophagy pathway. As a consequence, the autophagy deficient mutants, *atg5-1* and *atg7-2*, accumulate more endogenous FREE1 protein and display hypersensitivity to iron deficiency. Furthermore, under iron-deficient growth condition autophagy related genes are upregulated to promote the autophagic degradation of FREE1, thereby possibly relieving the repressive effect of FREE1 on iron absorption. Collectively, our findings demonstrate a unique regulatory mode of protein turnover of the ESCRT machinery through the autophagy pathway to respond to iron deficiency in plants.

## 1. Introduction

Owing to a complex living environment, plants are habitually faced with various hostile situations, especially from multifarious abiotic and biotic stresses. The process of plant adaptation is usually accompanied by the elimination of harmful cellular components like denatured proteins and damaged organelles. Therefore, plants have evolved a sophisticated degradation system to modulate the proper amount of essential proteins and to remove damaged cellular components [1]. One predominant protein degradation route is the ubiquitin-proteasome (UPS) system, which removes most of the improper and short-lived proteins in an ATP-dependent manner [2]. Moreover, the vacuolar system is also an important and conserved pathway which is rich in acid hydrolases with functions in protein degradation. These hydrolases inside the vacuole degrade proteins and organelles which are transported to the vacuole through various membrane trafficking pathways [3,4]. For example, cytoplasmic components like protein aggregates or damaged organelles can be enveloped by a double-membrane structure termed the autophagosome and delivered to the vacuole through the macroautophagy pathway (hereafter referred to as autophagy) [5,6,7]. The external membrane of the autophagosome fuses with the tonoplast, whereby the cargoes enveloped by the inner membrane are released as the autophagic bodies to the vacuolar lumen, then the inside cargo is degraded by various hydrolases for recycling [1,8].

Besides autophagy, the endosomal sorting complex required for transport (ESCRT) complex also regulates protein sorting to the vacuole, which is significant to multivesicular body (MVB) biogenesis and MVB-mediated vacuolar transport pathway [9,10]. Cell membrane proteins like the receptors and transporters are internalized into vesicles and delivered to the trans-Golgi network (TGN), which is defined as the early endosome in plants [11,12]. Depending on the demand, some TGN-localized proteins are recycled back to the cell membrane, while others are further transported to the late endosome, MVB, where those degraded membrane proteins are sorted to the intraluminal vesicles (ILVs) inside the MVB. Finally, the ILVs are released to the vacuolar lumen for degradation upon the fusion between the MVB and the vacuole [12,13]. During this process, the ESCRT complex plays essential roles in the sorting of ubiquitinated membrane proteins and the deformation of the MVB membrane to form ILVs [14,15]. Emerging evidence indicates that plants have evolved some unique ESCRT components, whose protein sequences are not very conserved among plants, yeast and mammals, but are essential for the plant ESCRT pathway [9,14]. One example is FYVE DOMAIN PROTEIN REQUIRED FOR ENDOSOMAL SORTING 1 (FREE1), which is incorporated into the ESCRT machinery through interacting with several important ESCRT components, such as the vacuolar protein sorting-associated protein 23 (Vps23), the sucrose nonfermenting protein 7 (Snf7) and the BRO1-LIKE DOMAIN-CONTAINING PROTEIN/APOPTOSIS-LINKED GENE-2 INTERACTING PROTEIN X (AtBRO1/ALIX), to regulate MVB-mediated endosomal sorting and vacuolar transport [16,17,18]. Besides the endosomal function in the cytoplasm, FREE1 also functions as a negative regulator to attenuate ABA signaling at the transcriptional level in the nucleus through phosphorylation-dependent shuttling from the cytoplasm to the nucleus [19]. Moreover, FREE1 also interacts with an autophagy regulator, SH3 DOMAIN CONTAINING PROTEIN2 (SH3P2), to modulate autophagic degradation [20,21]. Knockdown of *FREE1* causes accumulation of autophagosomes and blockage of autophagic degradation, possibly by affecting the fusion between autophagosomes and vacuoles in plant cells [21].

As an essential nutrient, iron affects substantial cellular functions, such as chlorophyll biosynthesis, photosynthesis and various redox reactions [22]. Therefore, the cellular content of iron in plants must be in a dynamic balance, to ensure there is enough iron participating in metabolism and to avoid the cytotoxicity caused by excessive iron [23,24]. Iron transporters located in the cell membrane play an important role in the iron absorption process and one of the most well-studied iron transporters is the IRT1 [25,26]. Plants alter the absorption of iron not only through affecting the transcript and protein abundance of IRT1, but also through affecting the dynamic localization of IRT1 [27,28,29]. Interestingly, previous studies have shown that FREE1 can function as a negative regulator to control IRT1-dependent iron absorption, possibly through modulating the dynamic localization of IRT1 to the plasma membrane [30,31]. Overexpression of FREE1/FYVE1 results in the apolar localization of IRT1 on the plasma membrane and the decrease of iron absorption in plants under iron-deficient growth conditions [29]. Recently, our and other studies uncovered the regulatory mechanism of FREE1 homeostasis mediated by the RING finger E3 ubiquitin ligases, SINA of Arabidopsis thaliana (SINATs) [31,32]. The ubiquitination and degradation of FREE1 are positively regulated by SINAT1–4 and negatively influenced by SINAT5 in response to iron-deficiency [31]. However, it remains unclear which degradation pathway (UPS or vacuole) the ubiquitinated FREE1 is going to degrade to respond iron-deficiency in plants.

In this study, we extended our previous study and tracked the degradation profile of the FREE1 protein in response to iron-deficiency. Using a combination of cellular and genetic approaches, we demonstrated that autophagy activity was enhanced to mediate the transport of ubiquitinated FREE1 protein to the vacuole for degradation when plants were challenged by iron-deficient growth condition. Our results uncover a unique regulatory mode of protein turnover of the ESCRT machinery through the autophagy pathway to respond to iron-deficiency in plants.

## 2. Results

### 2.1. FREE1 Can Be Degraded through an Autophagy-Dependent Pathway

Our previous research and other studies showed that SINAT1-4 E3 ligases regulate the ubiquitination and degradation of FREE1 [31,32]. To further explore the nature of FREE1 degradation, we performed a cycloheximide (CHX) treatment to chase the half-life and degradation profile of FREE1 protein. The obtained results showed that a reduction of FREE1 protein levels could be observed 16 h after CHX treatment and was more obvious after 24 h, indicating that FREE1 is a relatively stable and long-lived protein (Figure 1A). Moreover, the 26S proteasome inhibitor MG132 could partially retard FREE1 degradation, whereas the vacuolar H^+^-ATPase inhibitor concanamycin A (Conc A) could significantly block the decrease of FREE1 protein level upon CHX treatment (Figure 1A,D). These results indicate that FREE1 is degraded mainly through a vacuole-dependent mechanism, although we cannot fully rule out the partial contribution of the 26S proteasome in FREE1 degradation.

FREE1 has been shown to associate with SH3P2 and ATG8 to modulate autophagic degradation, possibly by affecting the fusion between autophagosomes and vacuoles [20,21,33]. Together with the above data showing vacuolar degradation of FREE1, we were thus motivated to test if FREE1 was degraded via the autophagy pathway by tracing the degradation profile of the FREE1 protein in two representative *Arabidopsis* mutants, *atg5-1* and *atg7-2*, which have been well documented as autophagy-defective lines due to the lack of ATG8 lipidation [34,35]. The obtained results showed that the degradation of FREE1 was largely abolished in *atg5-1* and *atg7-2,* even after CHX treatment for 24 h, indicating that autophagy is required for the degradation of FREE1 (Figure 1B,D). Moreover, consistent with our previous results showing the critical function of SINAT1, −2, −3 and −4 in mediating the ubiquitination and degradation of FREE1, the reduction of FREE1 protein level was significantly retarded in *sinat1234* quadruple mutants [31] (Figure 1C,D). Collectively, these data strongly suggest that FREE1 can be degraded through a SINATs-regulated and autophagy-dependent pathway.

### 2.2. Autophagy Regulates FREE1 Turnover and Plant Metal Homeostasis to Respond Iron Deficiency

Overexpression of FREE1 has been shown to result in plant hypersensitivity to iron- deficiency, since FREE1 is a negative regulator of IRT1-dependent metal absorption in plants [29]. Together with the above result showing that FREE1 is degraded mainly through an autophagy-dependent pathway, it raises a possibility that the autophagy pathway might be involved in FREE1-regulated plant metal homeostasis. To test this hypothesis, we first analyzed the endogenous protein levels of FREE1 in autophagy-deficient *Arabidopsis* mutants and found that both *atg5-1* and *atg7-2* had significantly increased FREE1 protein levels when compared with the wild-type plant (Figure 2A), a result consistent with the retarded onset of FREE1 protein degradation in these two *atg* mutants as shown in Figure 1B. Quantitative RT-PCR analysis revealed no obvious change in the level of *FREE1* transcripts among those plants (Figure 2B), indicating that the change in the level of FREE1 protein in *atg5-1* and *atg7-2* mutants was not due to the difference of *FREE1* expression. Moreover, we found that when compared with the wild-type plants, the *atg5-1* and *atg7-2* mutants showed hypersensitive phenotypes to iron-deficiency (Figure 2C,D), which were reminiscent of the phenotype of FREE1 overexpression plants [31]. In addition, the contents of Fe, Mn and Zn in *atg5-1* and *atg7-2* mutants grown in an iron-deficient medium were significantly less than those in the wild-type plants (Figure 2E), suggesting that autophagy might be engaged in the regulation of plant metal homeostasis, possibly through the modulation and the degradation of FREE1. Together, these data suggest that the autophagy pathway can regulate FREE1 turnover and metal homeostasis to respond to iron-deficient conditions.

### 2.3. Autophagy Positively Regulates the Degradation of Ubiquitinated FREE1 in Response to Iron Deficiency

To further clarify the regulation mechanisms of the autophagy-dependent FREE1 degradation in plant iron-deficiency responses, we examined the ubiquitination level of FREE1 protein in wild-type plants and *atg7-2* mutants. As shown in Figure 3A, the ubiquitination level of FREE1 protein was enhanced in the wild-type plants treated with the iron-deficient medium and accumulation of ubiquitinated FREE1 was much more obvious in *atg7-2* mutants. In addition, iron-deficiency conditions could obviously promote the degradation of FREE1 (Figure 3B); whereas such an effect was greatly compromised in autophagy mutants *atg5-1* and *atg7-2* (Figure 3C,D). These data suggested that iron depletion triggered the ubiquitination and the subsequent autophagy-dependent degradation of FREE1.

Vacuolar delivery of GFP fusion proteins always produces a characteristic vacuolar degradation product, termed GFP core [36]. By performing immunoblot analysis in *Arabidopsis* transgenic plants over expressing GFP-FREE1, we could detect an obvious accumulation of GFP core upon extended (3 day) iron-deficient treatment (Figure 3E), indicating vacuole-mediated degradation of GFP-FREE1. Reinforcing the conclusion of the autophagy-regulated vacuolar degradation of FREE1, treatment of GFP-FREE1 plants with iron deficient medium containing the vacuolar H^+^-ATPase inhibitor Conc A resulted in a dramatic accumulation of GFP-FREE1-decorated punctate foci within the lumen of vacuole (Figure 4). Compared to the wild type, the *atg7-2* plant exhibited no or fever GFP-FREE1 punctate foci inside the vacuolar lumen under iron-deficient conditions (Figure 4B,C). Furthermore, the serial section confocal scanning images confirmed the vacuolar localization of GFP-FREE1 puncta as well as its localizations on MVB puncta and nucleus as shown previously (Figure 4C) [17,19].

To confirm whether the vacuolar localized GFP-FREE1 puncta are autophagic bodies, we crossed the *GFP-FREE1* transgenic plant with a plant expressing the well-documented autophagosome marker mCherry-ATG8f for confocal observation [36,37]. The obtained results showed that upon iron starvation and Conc A treatment, these GFP-FREE1 labeled puncta inside vacuoles were largely overlapped with autophagic bodies marked by mCherry-ATG8f, as revealed by quantifying colocalization by the Pearson correlation coefficient (Figure 5A,B). Together, these results suggest that autophagy positively regulates the degradation of ubiquitinated FREE1 in response to iron-deficiency.

### 2.4. Autophagy Dynamically Responses to Iron Deficiency

To answer how plants respond to iron-deficiency, we examined the gene expression profiles of the representative *AUTOPHAGY-RELATED GENES* (*ATGs*) and the autophagy flux in the wild-type plants exposed to iron depletion. The expression levels of *ATG4a*, *ATG4b*, *ATG6*, *ATG8a* and *ATG8b* were continuously up-regulated and peaked 3 days after the transfer of plants to the iron-deficient medium (Figure 6A). Consistent with the induced gene expression of *ATGs*, we also found that iron starvation treatments dramatically enhanced autophagic flux, as revealed by the increased GFP/GFP-ATG8a ratio when quantified through their grayscale intensity in immunoblotting data (Figure 6B). Together with our previous results showing that iron depletion promotes the SINATs E3 ligase-regulated ubiquitination of FREE1 [31], these data suggest that iron depletion also activates the autophagy pathway to promote the protein turnover of ubiquitinated FREE1. Such iron-deficiency-induced upregulation of *ATGs* genes and enhancement of FREE1 ubiquitination and degradation might constitute a subtle regulation network to modulate plant iron-deficiency responses at both the transcriptional and the post-transcriptional layers.

## 3. Discussion

In this study, we found that the ESCRT component FREE1 could be degraded through the autophagy pathway. In previous studies, we and others showed that SINATs E3 ligases could modulate the ubiquitination and degradation of FREE1 protein to relieve its inhibitory effect on metal absorption or to adapt to other stressful growth conditions like plant recovery after ABA treatment [29,31,32]. In this study, we demonstrated an important and positive function of autophagy in mediating FREE1 degradation to respond to iron-deficiency. Previous studies have demonstrated the functions of autophagy in increasing zinc bioavailability under zinc deficiency and re-translocating nutrients like nitrogen and iron from vegetative organs to seeds [38,39]. Together with our current study, these results further reinforce the critical role of autophagy in plant adaptation to nutrient deficient conditions, where autophagy not only directly affects the re-translocation of nutrients in different plant tissues, but also possibly modulates the protein abundance of factors involved in the regulation of nutrient absorption or translocation.

Endocytosis and autophagy are well known as two major vesicular processes converging on the lysosome or vacuole and extensive attention has been drawn to explore the interplay between these two processes. Strikingly, several studies have demonstrated the involvement of the ESCRT machinery in the regulation of the autophagy pathway. For example, our previous work has revealed that the ESCRT component FREE1 directly interacts with SH3P2, a unique regulator in plant autophagy, to manipulate the autophagosome-vacuole fusion and autophagic degradation in plants [20,21]. Another study by Spitzer et al., demonstrates that the ESCRT-III accessory proteins CHARGED MULTIVESICULAR BODY PROTEIN1A (CHMP1A) and CHMP1B are required for autophagic turnover of plastids in *Arabidopsis* [40]. More recent work in yeast and mammalian cells have demonstrated that ESCRT machinery functions as a critical regulator of phagophore closure and autophagosome maturation [41,42,43]. These studies clearly demonstrate the function of ESCRT components in the autophagy pathway, but how the turnover of ESCRT components is regulated and especially whether autophagy can regulate ESCRT activity remains unknown. Several lines of evidence listed in this study demonstrate that autophagy could at least modulate the turnover of the ESCRT component FREE1: (1) CHX- or iron-deficiency-induced turnover of FREE1 is largely abolished in autophagy-defective mutants, (2) GFP-FREE1 is co-localized with autophagic marker mCherry-ATG8f in the vacuole under iron-deficient condition, (3) The *atg* mutants displayed similar phenotypes to that of FREE1 overexpression plants upon iron starvation treatment according to our previous report [31]. More work is needed to explore how the ubiquitinated FREE1 is captured by the selective autophagy pathway. One plausible receptor or mediator might be SH3P2, which possesses ubiquitin binding ability and can associate with both the ESCRT-I complex and ATG8 [20,21,44]. Another possibility is that FREE1 might be directly recognized by ATG8 as demonstrated in a recent study [33]. Moreover, it would also be interesting to investigate whether autophagy regulates other ESCRT-dependent cellular processes, such as MVB-mediated endosomal sorting and formation of viral replication complex inside host cells, under certain growth, or environmental response conditions.

Our current study showed that iron-deficiency could induce the autophagy pathway to decrease FREE1 abundance. This might act as a mechanism to let plants diminish FREE1, a negative regulator involved in iron absorption, and to make plants better adapt to the iron-deficient growth conditions, although we cannot rule out the possible implication of autophagy in the regulation of other metabolic changes and regulators involved in iron homeostasis under iron starvation conditions. Though we have shown that the expression of *ATGs* genes and the activity of autophagy were enhanced upon iron-deficiency, it remains unclear how iron-deficiency activate the autophagy pathway. One possible explanation is the enhanced expression of ATGs genes, which are the probable targets of iron-responsive transcription factors like FER-LIKE IRON-DEFICIENCY-INDUCED TRANSCRIPTION FACTOR (FIT) and UPSTREAM REGULATOR OF IRT1 (URI) as demonstrated previously [45,46]. Other potential explanations might include phosphorylation of ATGs proteins with subsequent activation of autophagy. More evidence come from previous reports showing that several kinases are engaged in the phosphorylation-dependent regulation of iron utilization [47]. It will be interesting to further dissect the interplay of the autophagy pathway and the signaling pathways involved in plant metal stress responses in future work.

## 4. Materials and Methods

### 4.1. Plant Materials and Growth Conditions

The Arabidopsis *atg5-1*, *atg7-2*, *sinat1234, p35S:GFP-ATG8a*, *pUBQ10:mCherry-ATG8f* and *pUBQ10:GFP-FREE1* were previously described [19,20,31]. The surface of seeds was sterilized with 15% sodium hypochlorite solution, soaked in water for 3 d at 4 °C and then sown on the medium plates with 1/2 MS (Caisson lab, MSP09-100LT), containing 1% sucrose and 0.8% agar. The plates were incubated for 7 d at 22 °C under long-day (LD) conditions (16 h light/8 h dark).

### 4.2. Iron Starvation-Related Phenotype Analysis

For the iron starvation-related phenotype assays, the method was handled as previously reported [29,31]. Plants were sterilized as described above and then cultivated on iron-sufficient 1/2 MS medium (containing 50 μM Fe-EDTA, marked as 1/2 MS + Fe) and iron-deficient 1/2 MS medium (containing 0 μM Fe-EDTA, marked as 1/2 MS − Fe). Pictures were acquired after normal growth for 7 d under LD conditions and Image J software was used to measure the roots length.

### 4.3. Metal Element Content Analysis

The method of measuring metal element content with ICP-MS was according to previous reports [29]. The seedlings were grown on 1/2 MS + Fe and 1/2 MS − Fe medium plates for 12 d and 0.5 g fresh-weight tissue were collected. The samples were first washed with 5 mM CaSO_4_ and 10 mM EDTA, then rinsed for 5 min with deionized water. Subsequently, the samples were digested with 5 mL 70% (*v*/*v*) HNO_3_ at 80 °C for 1 h, 100 °C for 1 h and 120 °C for 2 h by using a Microwave Digestion System (COOLPEX) after drying at 70 °C for 2 d. Finally, the treated samples were diluted 400 times with 3% (*v*/*v*) HNO_3_ and used to measure the contents of Fe, Mn and Zn with atomic ICP-MS System (Agilent 7800).

### 4.4. In Vivo Ubiquitination Assay

Transgenic plants expressing *pUBQ10:GFP-FREE1* in Col-0, *atg7-2* backgrounds were used to analyze the ubiquitination of FREE1 under the plant iron-deficiency condition. The 7-d-old transgenic plants were grown on 1/2 MS + Fe medium plates treated with 1/2 MS + Fe or 1/2 MS − Fe liquid medium for 1.5 d and collected the samples. Then total protein was extracted for immunoprecipitation with a GFP-trap (ChromoTek, gtma-20) as above described. Finally, the IP fractions were separated through SDS-PAGE and the ubiquitination of FREE1 could be analyzed by immunoblot using anti-GFP (Abcam, ab290), anti-Ub (Abcam, ab12930) antibodies.

### 4.5. Protein Extraction and Immunoblot Assay

Seven-d-old Col-0, *atg5-1*, *atg7-2* grown on 1/2 MS + Fe medium plates were harvested for total protein extraction to detect FREE1. Seven-d-old Col-0, *atg5-1*, *atg7-2* and *sinat1234* seedlings grown on 1/2 MS + Fe medium plates were treated with DMSO (Control), 100 μM CHX (cycloheximide, AbMole, M4879), 100 μM CHX + 50 μM MG132 (AbMole, M1902), 100 μM CHX + 0.5 μM Conc A (concanamycin A, AbMole, M7737) or iron starvation for the designed time points before harvesting to extract total protein for immunoblot analysis with anti-FREE1. Seven-d-old *p35S:GFP-ATG8a* seedlings were transferred from 1/2 MS + Fe medium plates to 1/2 MS − Fe liquid medium for 0 d, 1 d, 2 d and 3 d, respectively and harvested to assess the autophagy flux. All the plant samples were ground to powder in liquid nitrogen and fully homogenized with the 2 x SDS protein extraction buffer (0.5 M Tris-HCl, pH 6.8, 5% SDS, 20% glycerol and 10% β-Mercaptoethanol and 0.05% Bromophenol blue). Protein samples were boiled at 95 °C for 5 min before separation by SDS-PAGE and analyzed by immunoblot using anti-FREE1 [19,31], anti-actin (Abcam, ab197345), anti-GFP (Abcam, ab290) antibodies and anti-H3 (Abcam, ab1791), respectively.

### 4.6. Laser Scanning Confocal Microscope Observation

Six-d-old Arabidopsis seedlings expressing *pUBQ10:GFP-FREE1* in Col-0 and *atg7-2* backgrounds or the F1 plants of *pUBQ10:GFP-FREE1* (Col-0) crossed with *pUBQ10-mCherry-ATG8f* (Col-0) grown on 1/2 MS + Fe medium plates were transferred to 1/2 MS − Fe liquid medium for 12 h, then supplemented with 0.5 μM Conc A for another 8 h prior to imaging using a Zeiss 800 Laser scanning confocal microscope. GFP and mCherry were excited by a 488-nm and a 561-nm laser, respectively. Microscopy images were processed using the ZEN imaging software (Zeiss). The colocalization ratio of GFP-FREE1 and mCherry-ATG8f was quantified by using ImageJ software with a Pearson–Spearman correlation (PSC) plugin as described previously [48,49].

### 4.7. RNA Extraction and Gene Expression Analysis

Col-0, *atg5-1*, *atg7-2* was grown on 1/2 MS medium plates for 7 d, harvested and total RNA was extracted, to analyze the gene expression of *FREE1* by qRT-PCR. The autophagy related genes are measured by the use of the 7-d-old Col-0. The seeds were cultivated on 1/2 MS + Fe medium plates for 7 d and then transferred to 1/2 MS − Fe liquid medium, the samples were harvested respectively after 0 d, 1 d, 2 d and 3 d treatment as the previous designed time points. Total RNA could be extracted with the Hi-Pure Plant RNA Mini Kit (Magen, R4151). The first-strand cDNA was synthesized from 1 µg total RNA using the PrimeScriptTM RT reagent Kit with gDNA Eraser (TaKaRa, RR047Q). Quantitative real-time PCR was performed on the CFX96 Touch real time PCR detection system (BioRad) using TB Green^®^ Premix ExTaq™ II mix (TaKaRa, RR820Q). Relative abundance of target transcripts was determined by the 2^−ΔΔCT^ method, using the levels of Arabidopsis *Ubiquitin10* transcript as the internal control. The specific primers used in this study were listed in Appendix A.

### 4.8. Data Processing and Statistical Analysis

The root length, the band intensity of Western blots could be measured by Image J. and GraphPad Prism software was used to acquire the histograms according to at least three independent biological replicates. SPSS was adopted to analyze statistical differences with one-way ANOVA. Different letters indicate means that were statistically different by Tukey’s multiple testing methods (*p* < 0.05).

## Figures and Tables

**Figure 1 ijms-22-08779-f001:**
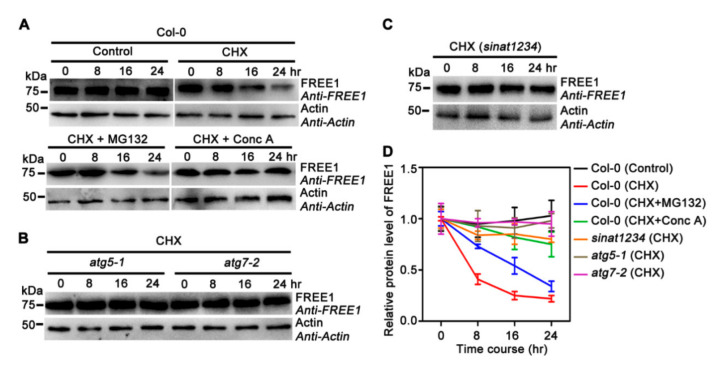
Autophagy-dependent degradation of FREE1. (**A**) Endogenous FREE1 levels in 7-d-old Col-0 seedlings upon DMSO (Control), 100 μM CHX, 100 μM CHX + 50 μM MG132 and 100 μM CHX + 0.5 μM Conc A treatments for 0, 8, 16 and 24 h. (**B**) The degradation profiles of FREE1 in *atg5-1* and *atg7-2* upon 100 μM CHX treatment for 0, 8, 16 and 24 h. (**C**) The degradation profile of FREE1 in *sinat1234* upon 100 μM CHX treatment for 0, 8, 16 and 24 h. (**D**) Quantification of the relative level of FREE1 protein under the indicated backgrounds or treatments. Values stand for the relative grayscale intensities of FREE1 normalized to actin and the first lane in each experiment was set as 1. Data represented as mean ± SD from three independent experiments.

**Figure 2 ijms-22-08779-f002:**
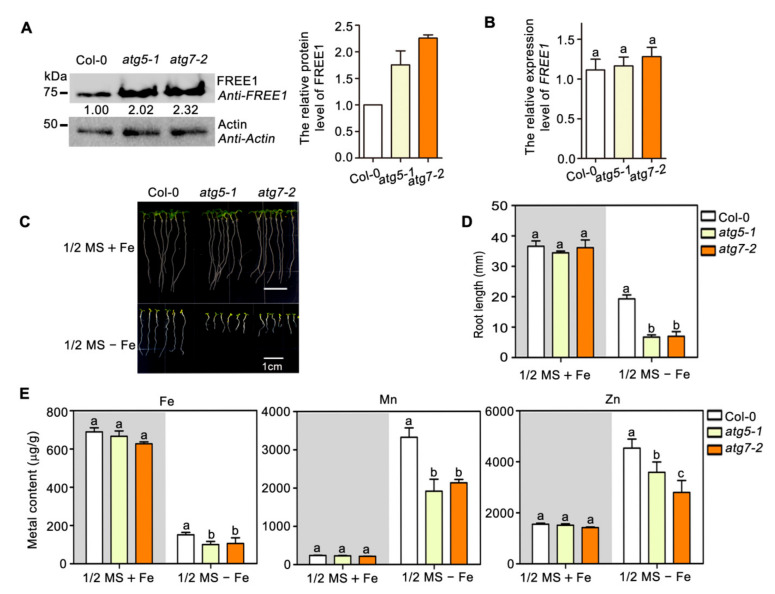
Autophagy positively regulates metal uptake under iron-deficient growth conditions. (**A**) Immunoblot-detection of endogenous FREE1 in 7-d-old Col-0, *atg5-1* and *atg7-2* seedlings. The numbers below the corresponding bands were relative grayscale intensities of FREE1 normalized to actin. Quantification of the relative grayscale intensities of FREE1 from three independent experiments was shown in the right panel. (**B**) qRT-PCR analysis of *FREE1* expression levels among the Col-0, *atg5-1* and atg7-2 mutants. *UBQ10* was used as an internal control for qRT-PCR analysis. (**C**) Phenotypes of Col-0, *atg5-1* and atg7-2 vertically grown for 7 d on 1/2 MS + Fe (upper panel) and 1/2 MS − Fe (lower panel) medium plates. (**D**) Root length in forementioned seedlings (**C**). Data are represented as mean ± SD of 20 seedlings from three replicates. (**E**) Metal content was determined by ICP-MS of 12-d-old seedlings grown on 1/2 MS + Fe and 1/2 MS − Fe medium plates. Results are presented as mean ± SD from three batches of seedlings, each batch weighing 0.5 g in fresh weight. Statistical differences were calculated by one-way ANOVA. Different letters indicate means that were statistically different by Tukey’s multiple testing methods (*p* < 0.05) for genotypes within a given growth condition.

**Figure 3 ijms-22-08779-f003:**
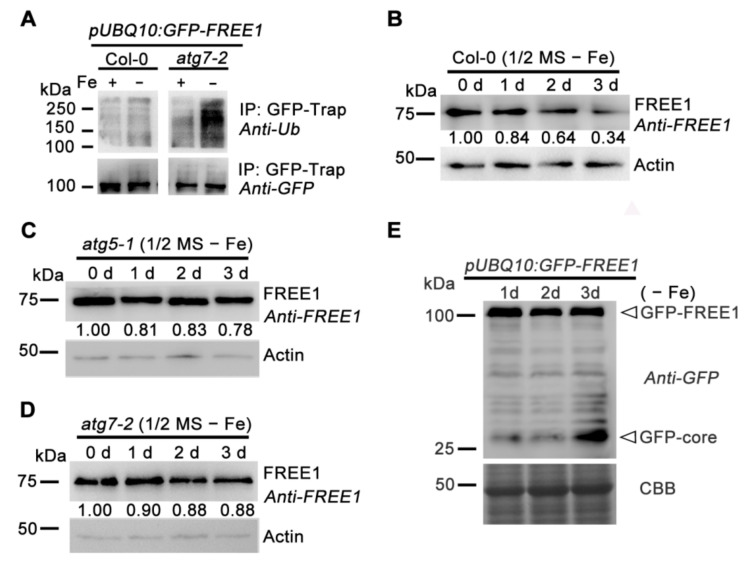
Autophagy positively regulates the degradation of FREE1 in response to iron-deficiency. (**A**) The total ubiquitination of GFP-FREE1 in response to iron starvation in Col-0, *atg7-2* backgrounds. Seven-d-old plants expressing *pUBQ10:GFP-FREE1* in the indicated backgrounds grown on 1/2 MS + Fe medium plates were treated with 1/2 MS + Fe or 1/2 MS − Fe liquid medium for 1.5 d before harvesting to perform protein extraction followed by immunoprecipitation (IP) with a GFP-trap. Finally, the IP fractions were subjected to immunoblot analysis with the indicated antibodies. (**B**–**D**) Endogenous FREE1 levels in 7-d-old Col-0, *atg5-1* and *atg7-2* seedlings upon iron starvation exposure for 0 d, 1 d, 2 d and 3 d, respectively. (**E**) The 5-d-old plants over expressing *pUBQ10:GFP-FREE1* upon iron starvation exposure for 1 d, 2 d and 3 d were used for protein extraction followed by detection with a GFP antibody.

**Figure 4 ijms-22-08779-f004:**
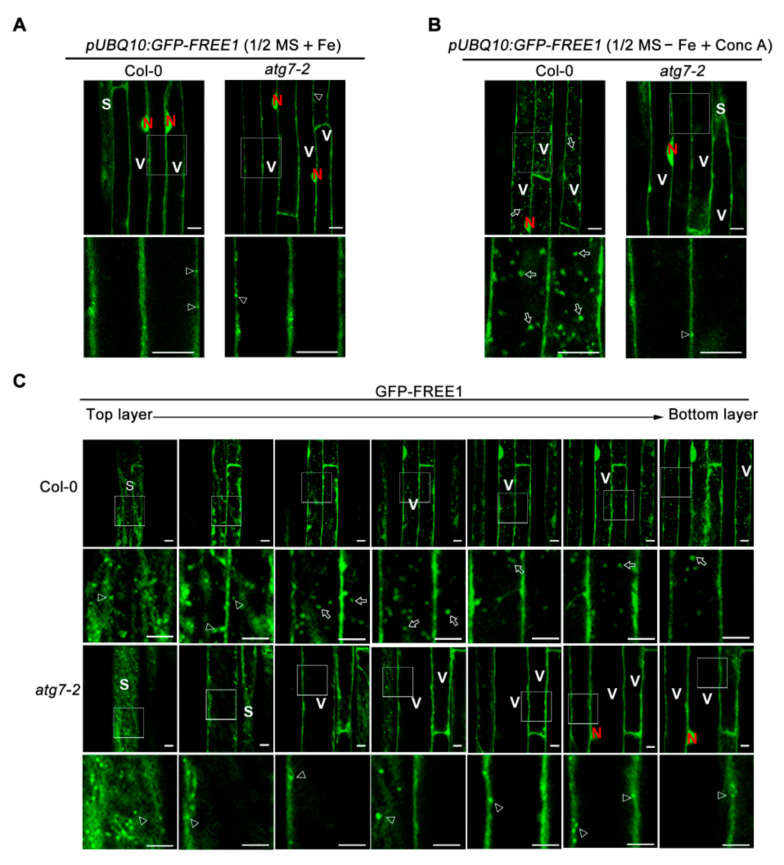
Autophagy-dependent FREE1 entry into the vacuole in response to iron-deficiency. (**A**) and (**B**) 6-d-old Arabidopsis seedlings expressing *p**roUBQ10:GFP-FREE1* in Col-0 and *atg7-2* background grown on 1/2 MS + Fe medium plates (**A**) were transferred to 1/2 MS-Fe liquid medium for 12 h followed by another 8 h treatment in 1/2 MS − Fe liquid medium containing 0.5 μM Conc A before confocal observation of the root elongation zone. The untreated plants were used as a control. The punctate MVB localization of GFP-FREE1 was easily observed in the surface layer (the thin cytoplasm region) of the root cells in the elongated zone. The number of GFP-FREE1 puncta inside vacuole was obviously decreased in *atg7-2* mutant comparing with that in WT plant cells. Bars = 10 μm. (**C**) Serial confocal optical section of those plants shown in (**B**). S stands for surface layer of the indicated cell; V stands for vacuole; N stands for nucleus. The punctate MVB localization of GFP-FREE1 could be seen in the surface layer (the thin cytoplasm region) of the root cells and was indicated by arrow heads. The representative GFP-FREE1 puncta inside vacuole was indicated by arrows. Bars = 10 μm.

**Figure 5 ijms-22-08779-f005:**
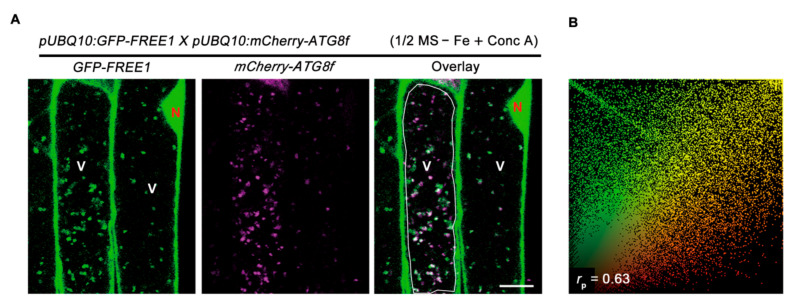
Colocalization of GFP-FREE1 with autophagic bodies inside vacuole. (**A**) Images of colocalization between GFP-FREE1 and mCherry-ATG8f in the lumen of vacuole. Six-d-old *p**roUBQ10:GFP-FREE1 x p**roUBQ10:mCherry-ATG8f* plants grown on 1/2 MS + Fe medium plates were transferred to 1/2 MS − Fe liquid medium with 0.5 μM Conc A treatment as shown above before confocal imaging of the root elongation zone. Bars = 10 μm. (**B**) Colocalization ratio of GFP-FREE1 and mCherry-ATG8f. Only vacuole regions indicated by white rectangle in (**A**) were used for quantification of colocalization ratio. Colocalization relationship indicated by *r*_p_ value was measured from five independent confocal images by using Image J software with Pearson–Spearman correlation plugin. The maximum *r*_p_ value is one standing for full colocalization.

**Figure 6 ijms-22-08779-f006:**
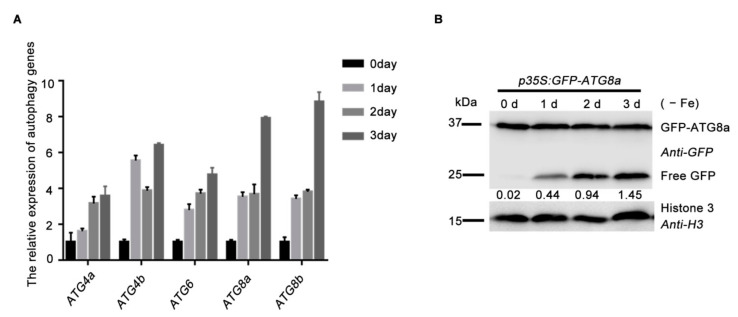
Iron-deficiency promotes the activity of autophagy. (**A**) Gene expression profile of ATGs in 7-d-old Clo-0 seedlings exposed to iron starvation for 0 d, 1 d, 2 d and 3 d. Overall gene expression levels of ATGs were increased. Data represent the mean ± SEM from three independent experiments. (**B**) Immunoblot analysis of autophagic flux in response to Fe starvation. Seven-d-old *GFP-ATG8a* seedlings grown on 1/2 MS + Fe medium plates were transferred to 1/2 MS − Fe liquid medium for treatment 0 d, 1 d, 2 d and 3 d followed by protein extraction and immunoblot analysis. Values below the blots represent the ratio of free GFP to GFP-ATG8a fusions in the individual lanes.

## Data Availability

Data is contained within the article or Appendix A.

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
