# Peer review of "Autophagy Mediates the Degradation of Plant ESCRT Component FREE1 in Response to Iron Deficiency"

_ijms, 2021, doi:10.3390/ijms22168779_

Round 1
Reviewer 1 Report
The manuscript of Zhang et al. concerns the mechanism of FREE1 degradation upon iron deficiency conditions in A. thaliana. Authors show an interesting regulation and impact of autophagy on the abundance of ESCRT-related protein. The research has been designed properly and obtained data are mostly clear and convincing. The one exception is RT-PCR result presented at Fig. 2b. It looks that the expression of FREE1 in atg5-1 and atg7-2 is slightly higher than in Col-0 plants and the authors conclude that it is not. I suggest to run the quantitative RT-PCR so that the result does not raise any doubts, as the assessment by an eye may always be imprecise.
Minor points to improve are:
- Fig. 2c – please add the scale.
- Please explain abbreviations SNF7 and AtBRO/ALIX when mentioned in the text.
- Fig. 4c – please show only stacked photographs and not so many single layers. It is unnecessary.
- Fig. 5 – please use magenta and green instead of red and green colors as more appropriate for the problem of color blindness of some readers.
Author Response
Please see our responses in the attached PDF file.

Reviewer 2 Report
It is a long-standing question if there is a direct involvement of FREE1 protein in autophagy pathway. Previous reports rectified if FREE1 knockdown jeopardize autophagy. FREE1-IRT1 interaction is well known and only link of FREE1 with IRT1 homeostasis.
Initially, the major evidences were toward FREE1 as master ESCRT component but dispensable for autophagy progression. It is indeed in interest of plant autophagy community if there are new evidence for FREE1 degradation by autophagy machinery.
In work by Zhang et al., author brings hypothesis of FREE1 regulation by autophagy to limit IRT1-FREE1 complex degradation through proteasome.
Following some comments those may improve the current form of MS.
Author should mention in introduction the autophagy flux fate in FREE1 RNAi plants as published in previous works
Authors should include WT in figure 1B and C
Quantification of signal in figure 2A with replicates, approximate 1-fold difference doesn’t seem too much to convenience
Figure 2b legend authors should mention number of cycles used for RT-PCR
I don’t understand how root/iron uptake phenotype justify the sole FREE1 involvement with autophagy and iron uptake. Since, autophagy deficient mutants could have plethora of metabolic changes and differential regulation of upstream regulators of iron homeostasis. Author should include proper controlled evidences for this like using FREE1-RNAi lines with their root length/iron uptake phenotypes.
Signal quantification is must for Figure 3B. Probably loading three different volumes 1x, 4x, 8x would make it more confirmatory, since it is one of the most important blot in hypothesis.
Authors should validate the flux of FREE1 autophagy degradation from usual endosomal and proteolytic pathway. They should test if there is less colocalization of GFP-FREE1 with endosomal marker under Fe starvation situation. Since the overall change in FREE1 level is not very dramatic under autophagy deficient mutants (figure 2A). Author should include FREE1-ATG8 colocalization with autophagy inducers like BTH/AZD as control to prove that the colocalization they are observing is Iron deficiency induced.
In discussion, authors should soften their claims specially when they are self-explaining that master regulators of Autophagy-iron homeostasis are still unexplored. While FREE1 might be a mild link, but this couldn’t be most potential candidate to understand this nexus. Specially the transcript level of ATGs genes doesn’t say much.
Author Response

(The authors gave the same response as above.)

Round 2
Reviewer 2 Report
The revised MS has answered all related concerns.